# Crossmodal clustered contrastive learning:
# Grounding of spoken language to gesture

DONG WON LEE, Carnegie Mellon University, USA

CHAITANYA AHUJA, Carnegie Mellon University, USA

LOUIS-PHILIPPE MORENCY, Carnegie Mellon University, USA

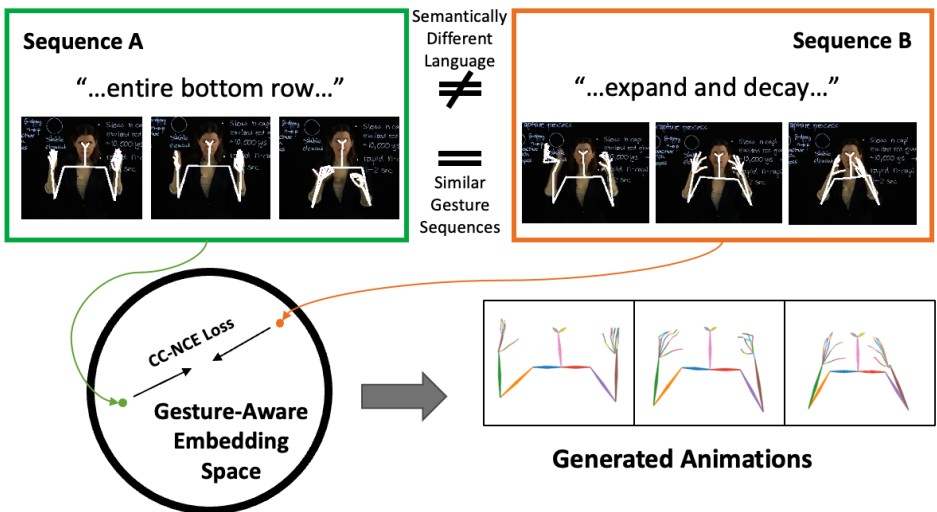

Fig. 1. Consider the two aligned sequences of spoken language phrases and gestures. The phrases, "entire bottom row" and "expand and decay" are semantically different. Hence, their respective language embeddings are far apart in the latent space. However, they are accompanied by the same gesture. Thus, we guide the embeddings to be closer in the gesture-aware embedding space, which is used for the downstream task of gesture generation.

Crossmodal grounding is a key challenge for the task of generating relevant and well-timed gestures from just spoken language as an input. Often, the same gesture can be accompanied by semantically different spoken language phrases which makes crossmodal grounding especially challenging. For example, a deictic gesture of spanning a region could co-occur with semantically different phrases "entire bottom row" (referring to a physical point) and "molecules expand and decay" (referring to a scientific phenomena). In this paper, we introduce a self-supervised approach to learn such many-to-one grounding relationships between spoken language and gestures. As part of this approach, we propose a new contrastive loss function, Crossmodal Cluster NCE , that guides the model

Authors' addresses: Dong Won Lee, Carnegie Mellon University, 5000 Forbes Avenue, Pittsburgh, PA, USA, dongwonl@andrew.cmu.edu; Chaitanya Ahuja, Carnegie Mellon University, 5000 Forbes Avenue, Pittsburgh, USA, cahuja@andrew.cmu.edu; Louis-Philippe Morency, Carnegie Mellon University, 5000 Forbes Avenue, Pittsburgh, PA, USA, morency@cs.cmu.edu.

to learn spoken language representations which are consistent with the similarities in the gesture space. By doing so, we impose a greater level of grounding between spoken language and gestures in the model. We demonstrate the effectiveness of our approach on a publicly available dataset through quantitative and qualitative studies. Our proposed methodology significantly outperforms prior approaches for grounding gestures to language. Link to code: https://github.com/dondongwon/CC_NCE_GENEA.

CCS Concepts: • **Computing methodologies** → **Learning latent representations**; **Neural networks**; **Natural language processing**.

Additional Key Words and Phrases: Gesture generation; virtual agents; socially intelligent systems; co-speech gestures; multi-modal interaction; contrastive learning; crossmodal translation; deep learning

## 1 INTRODUCTION

Nonverbal behaviours such as body posture, hand gestures and head nods play a crucial role in human communication [42]. Pointing at different objects, moving hands up-down in emphasis, and describing the outline of a shape are some of the many gestures that co-occur with the verbal and vocal content of communication. The language content, including spoken words (verbal cues) are co-generated with gestures to express meaning [21, 31]. When creating new robots or embodied virtual assistants designed to communicate with humans, it is important to generate gestures that are *relevant* with language and speech [5, 22, 34].

Imagine a person gesturing erratically, waving their arms in a way that is unrelated to what they are saying. Even in human-to-human conversation, this interaction would be considered unnatural. Similarly, a robot generating irrelevant gestures is a huge concern, as the wrong accompaniment of gestures could make humans uncomfortable interacting with the robot. Some previous works have focused on the coverage and diversity of the gestures [1, 44]. In this work, we primarily focus on the precision of the generated gestures. Hence, we need to enforce greater levels of grounding, so that the generated gestures are more relevant with the language. In a sense, we want the robot to be more cautious of generating erratic gestures. A way to tackle this challenge is via restricting the mapping of semantically different language to a smaller subset of high quality gestures.

Consider a person saying 'Someone gave *me* a gift yesterday' and '*My heart* is beating is so fast'. The deictic gesture of pointing at themselves is likely to co-occur with the spoken word '*me*' as well as '*my heart*'. Notice how the spoken words and meanings are very different, however, the accompanying gestures are quite similar. This sheds light on the existence of many-to-one relationship between spoken language and gestures [12, 35] and modelling this relationship is a key technical challenge. Specifically, solely relying on a reconstruction loss to learn crossmodal grounding can imply that the grounding relationships are one-to-one. However, at times, the true relationship between spoken language and gestures may be many-to-one.

Specifically, given two semantically different language sequences, their latent language representations must be close together if their accompanying gestures are similar. In order to address this problem, the key challenge is to guide the language latent to be aware of similarities and dissimilarities in the gesture space. We introduce the Crossmodal Cluster Noise Contrastive Estimation (CC-NCE) objective to learn a gesture-aware embedding space, where the similarities and disimilarities of samples in the gesture space are preserved. Our loss guides the model to learn a gesture-aware embedding space, where spoken language representations are consistent with the intra-cluster similarities and inter-cluster dissimilarities in the gesture space. In order to do so, we construct clusters in the output space of gestures with a new self-supervised mechanism. This provides positive and negative samples for many-to-one grounding, which is a key challenge as it requires additional knowledge of the output gesture modality. Also, the construction of unsupervised

Manuscript submitted to ACM

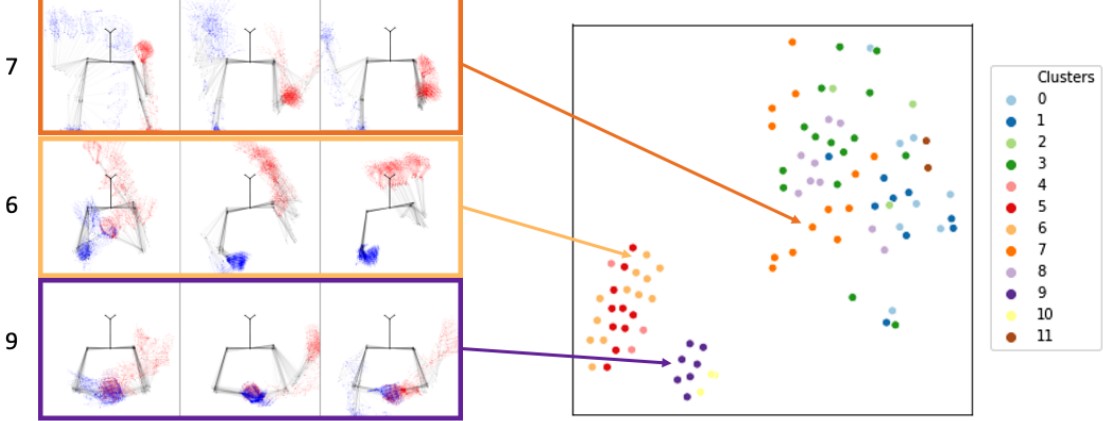

Fig. 2. The heatmaps on the left demonstrate intracluster similarity and intracluster dissimilarity of gestures clustered by our algorithm in a self-supervised manner. On the right, the t-SNE plot of our gesture-aware langauge embeddings, $Z$, demonstrate that our proposed Crossmodal Cluster Noise Contrastive Estimation brings together spoken language embeddings for similar gestures.

clusters can be computationally heavy for large datasets and requires the number of clusters which comes at a cost of an additional hyperparameter. To combat these technical challenges, we propose an online approach for constructing these clusters where the number of clusters are dynamically chosen while learning the crossmodal translation model.

Our proposed CC-NCE Loss places an emphasis in learning the many-to-one grounding between language and gesture. This serves as a method to prevent erratic co-speech gestures, which could interfere with natural human-computer interaction. Therefore, we focus on the precision of the generated gesture sequences. We conduct our experiments on the publicly available PATS dataset [2]. We find that CC-NCE provides additional incentive for the model to generate a smaller subset of higher quality gestures closer to the ground truth, with better performance on accuracy metrics. However, we also perceive the effects of precision-coverage trade off, where the emphasis in precision and grounding comes at a cost of a decrease in the coverage metrics.

## 2 RELATED WORKS

*Language in Gesture Generation.* A rule-based approach was proposed in an earlier study by Cassell et al. [6], where the behavior expression animation toolkit (BEAT) was developed to schedule behaviors, such as hand gestures, head nods and gaze. This approach was extended to utilize linguistic information from input text for selecting rules. [24, 27, 28, 30, 43].

Rule based approaches were replaced by deep conditional neural fields [8, 9] and Hidden Markov Models for prosody-driven head motion generation [38] and body motion generation [25, 26]. These use a dictionary of predefined animations, limiting the diversity of generated gestures. Soon, neural network based models were introduced, using unimodal inputs, specifically speech, to generate a sequence of gestures [18], head motions [37] and body motions [2, 3, 13, 14, 40]. On the other hand, Yoon et al. [45] uses only a text input for gesture generation. More recently, multimodal models utilizing both speech and language were developed. Kucherenko et al. [23] combines the two representations via early fusion. In order to account for the bi-modal relationship between language and audio in the

input modalities, Ahuja et al. [1] utilizes a crossmodal attention mechanism to account for correlations between speech and language. It is important to note that prior approaches [1, 2, 14, 23, 44] rely solely on reconstruction losses (L1 distance between generated pose and ground truth) to learn the grounding between gestures and language. In this paper, we argue that the inclusion of an additional contrastive grounding loss is valuable to the model, specifically to learn the many-to-one mapping between spoken language and gestures.

*Contrastive Learning.* Contrastive learning has gained traction recently due to its success in self-supervised learning. Oord et al. [32] intially proposed the Contrastive Predictive Coding method to learn informative representations in a self-supervised manner via Noise Contrastive Estimation (NCE). NCE primarily relies on learning an parametrized encoder to estimate the true distribution (positives) against random noise (negatives). He et al. [19] proposed MoCo, which stores a long queue of samples, to insert as negatives to contrast with augmented anchor samples. Chen et al. [7] proposed SimCLR, which utilized large batch sizes, and eliminating the need for large stored dictionaries. Park et al. [33] offered a methodology called Patch-wise contrastive Loss, which maximizes the mutual information between corresponding input and output patches. More recently, a vein integrating clustering mechanisms with contrastive learning has been proposed, where unsupervised clusters are built in a unimodal space and noise contrastive estimation is applied [4, 20, 29, 39]. Finally, pertinent to our crossmodal task, Udandarao et al. [41] projects each modality into a joint embedding space where both modalities are present. Then, they used supervised labels to retain intra-class and interclass relationships for clusters in the joint space. Furthermore, their methods are designed for downstream discriminative tasks, whereas our task is generative. A key distinction is that our work utilizes self-supervision to construct clusters, specifically in the output modality. We utilize the clusters in the output modality such that the same nature in is preserved in the representations of the input modality.

## 3 GESTURE GENERATION PROBLEM

Our primary task is to learn a generative model which translates language (BERT [11] tokens) and speech (log-mel spectrograms) modalities to relevant co-speech gestures. To that end, we learn a joint embedding space where sentences $\mathbf{X}^w$ and speech signals $\mathbf{X}^a$ are mapped to latent embeddings $\mathbf{Z} \in \mathcal{Z}$ using an encoder $G_e$. These latent embeddings are further mapped to the space of human upper-body poses represented in temporal skeletal keypoints, (i.e $\hat{\mathbf{Y}}^p$) using a decoder $G_d$ to optimize for the downstream task of gesture generation.

Formally, we are given a sentence of $K$ language tokens $\mathbf{X}^w = \left[x_1^w, x_2^w, \ldots x_K^w\right]$ and a sequence of co-occuring speech features, $\mathbf{X}^a = \left[x_1^a, x_2^a, \ldots x_T^a\right]$. We want to predict a sequence of T gesture poses $\mathbf{Y}^p = \left[y_1^p, y_2^p, \ldots y_T^p - 1\right]$ with $\mathbf{X}^a$ and $\mathbf{X}^w$ as input. Here $y_t^p \in \mathcal{R}^{J \times 2}$ are the xy-coordinates for $t^{th}$ frame for $J$ joints of the body skeleton. $x_t^w \in \mathcal{R}^{\mathcal{K}}$ is the $t^{th}$ word embedding with dimension $K$ and $x_t^a \in \mathcal{R}^{\mathcal{M}}$ is the $t^{th}$ speech frame with dimension $M$.

Thus, we have,

$$\mathbf{Z} = G_e(\mathbf{X}^a, \mathbf{X}^w; \theta) \tag{1}$$

$$\hat{\mathbf{Y}}^p = G_d(\mathbf{Z}; \psi) \tag{2}$$

Parameters of this encoder-decoder model, $\theta, \psi$, are optimised with true poses $\mathbf{Y}^p$ as a training signal, which can be written as a reconstruction loss, $L_{rec}(\theta)$ where we use the following L1 distance based on prior works [1, 2, 14, 23, 44, 45],

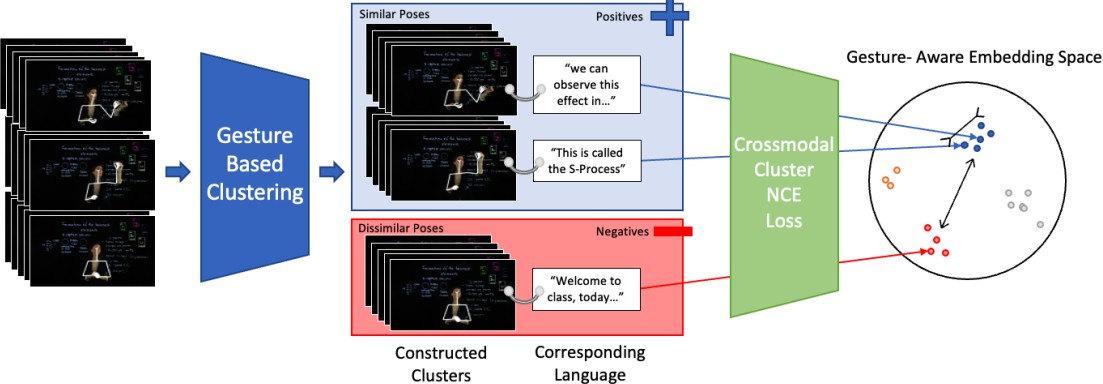

Fig. 3. Our proposed approach of self-supervised clustering in the output space of gestures, then utilizing the constructed clusters to sample negative and positives for the Crossmodal Cluster NCE loss to learn a gesture-aware language embedding space.

$$\mathcal{L}_{rec}(\theta, \psi) = \mathbb{E}_{\mathbf{Y}^p, \mathbf{X}^a, \mathbf{X}^w} \| \mathbf{Y}^p - G_d(G_e(\mathbf{X}^a, \mathbf{X}^w))) \|_1. \tag{3}$$

Often, as in GAN-based models [1, 14], adversarial losses [15] are included to alleviate the challenge of overly smooth generation and regression to the mean caused by reconstruction loss [14]. This adversarial loss is written as:

$$
\begin{aligned}
\mathcal{L}_{adv}(\theta, \psi, \eta) \quad = \quad & \mathbb{E}_{\mathbf{Y}^p} \log D_\eta \left( \mathbf{Y}^p \right) \\
& + \mathbb{E}_{\mathbf{X}^a, \mathbf{X}^w} \log \left( 1 - D_\eta (G_d(G_e \left( \mathbf{X}^a, \mathbf{X}^w \right))) \right)
\end{aligned}
\tag{4}
$$

The model is jointly trained to optimize the overall loss function $\mathcal{L}$,

$$\max_{\eta} \min_{\theta, \psi} \mathcal{L}_{rec}(\theta, \psi) + \lambda \mathcal{L}_{adv}(\theta, \psi, \eta) \tag{5}$$

The above formulation is similar to previous works in gesture generation [1, 23, 45].

## 4 METHOD

Our key contribution in this paper is to explicitly model the many-to-one mapping between spoken language and gestures in the latent space. This approach involves a two-step process, as shown in Figure 3. Our novel loss function $L_{cc-nce}$ guides the aligned language representations to be close to each other if their corresponding ground truth gestures are in the same cluster, and far apart if their corresponding ground gestures gestures are not in the same cluster. Thereby, creating a gesture-aware embedding space. We also propose a clustering algorithm to find similar gestures in a self-supervised, online manner. The algorithm first constructs batch-wise clusters, which compares itself with the global clusters and then decides whether the batch-level cluster should be merged or form its own cluster.

Finally, the optimization of the combined objective function describes the full model,

$$\max_{\eta} \min_{\theta, \psi} \mathcal{L}_{rec}(\theta, \psi) + \mathcal{L}_{adv}(\theta, \psi, \eta) + \mathcal{L}_{cc-nce}(\theta, \psi) \tag{6}$$

## 4.1 Crossmodal Cluster NCE

Given the same gesture, many different spoken phrases could accompany it, as shown in Figure 1. Therefore, even semantically different language embeddings corresponding to similar gesture sequences should be mapped closer together in the latent space. In order to do so, we propose the Crossmodal Cluster Noise Contrastive Estimation Loss, inspired by the InfoNCE Loss [16, 17, 32] to learn the gesture-aware embedding space.

*4.1.1 Gesture-aware Embedding Space.* The InfoNCE Loss [16, 17, 32] first samples an anchor sequence. Its augmentations are considered as positive samples, whereas the remaining elements within the batch (or a stored queue) are considered as negative samples [7, 19]. We want to guide the language latent space to be close together for similar gesture sequences and far apart from other dissimilar ones. Hence, sampling a positive or negative sample from the dataset requires additional knowledge of the output gesture modality. To tackle this challenge, we construct unsupervised clusters in the output gesture modality, which is described in the next section.

With these constructed clusters in the gesture-domain, we want to coerce the corresponding language embeddings to mimic the inter-cluster and intra-cluster relationships in the gesture space. We are given an anchor with ground truth gesture sequences and the corresponding language embeddings, $[y, z]$ respectively. We are also given global clusters of gesture sequences and their corresponding language embeddings. At this step, we want to find the cluster which contains gesture sequences that are most similar to the anchor gesture sequence. Mathematically, given a set of clusters $C$, we find the most similar gesture sequence and the aligned language embeddings: $y_c^+, z_c^+ = argmax\ (Sim(y_c, y))$, $\forall$ $[y_c, z_c] \in C$. Given the anchor, $z$, we use the corresponding language embeddings of the most gesture-wise similar cluster as the positive samples $z_c^+$. The language embedding sequences in other clusters will be considered as negative samples $z_c^- = [C \backslash z_c^+]$. With this assignment, we utilize properly assigned samples, in our Crossmodal Cluster NCE .

$$L_{cc-nce} = -\mathbb{E}_z \left[ \log \frac{\exp(F(z)^T F(z_c^+))}{\exp(F(z)^T F(z_c^+)) + \exp(F(z)^T F(z_c^-))} \right] \tag{7}$$

The Crossmodal Cluster NCE as shown in Equation 7 guides the language embedding space to learn the similarities in the output domain and projects them into a gesture-aware embedding space. The numerator encourages the semantically different language representations to be closer since they belong in the same gesture cluster. Given an anchor sequence $z$, and gesture-wise similar positive language embeddings $z_c^+$ and their dissimilar negatives $z_c^-$, we feed these language embeddings into an encoder, which we denote as $F(.)$ to learn the relationships in the gesture space.

*4.1.2 Gesture-Based Clustering.* We want to embed the knowledge of the many-to-one relationship between spoken language and gestures as shown in Figure 1. To do so, we need to find clusters of similar gestures to provide positive and negative samples for many-to-one grounding. Since we are not provided with annotations of similar gesture clusters, we must do this in a self-supervised way.

The construction of unsupervised clusters can be computationally heavy for large datasets and requires the number of clusters which comes at a cost of an additional hyperparameter. To combat these technical challenges, we propose an online approach for constructing these clusters where the number of clusters are dynamically chosen while learning the crossmodal translation model.

We iterate through the data and find the mean $\hat{\mu}$ and standard deviation $\hat{\sigma}$ of the pairwise dot-product similarity (referred to as $Sim$) of two arbitrary sequences of gestures. This metric is updated using a moving average continuously. These metrics are added and used as a threshold to find whether two sequences are similar or not. For example, a sequence $x$ is deemed similar to $y$, if $Sim(x, y) \geq \hat{\mu} + \hat{\sigma}$.

In practice, constructing and utilizing gesture-based clusters in an online manner is a two step process, (1) Batch Clustering and (2) Global Clustering, which is discussed below.

*(1) Batch Clustering:* The construction of the batch-wise cluster is important, as we can only compute the gradients with respect to the batch-wise embeddings and it would be infeasible to work with the global clusters due to computational limitations. We construct these clusters to utilize as anchor sequences $z$.

We describe the algorithm that is used to find the batch-level gesture clusters. In the first step, we calculate similarity metrics for an arbitrarily chosen anchor pose sequence, $y_a^b$, with the other pose sequences in the batch, $y^b[\sim L]$, where "$\sim L$" are indices of sequences in the batch which has not been assigned to a cluster yet. The anchor sequence and sequences in the batch, which yield a similarity score greater than the threshold $(Sim(y_a^b, y^b[\sim L]) \geq \hat{\mu} + \hat{\sigma})$, are assigned to a batch-wise cluster. Within the batch-level clustering, we want to discover clusters that are very different from each other. By assigning the next anchor sequence to the sequence with the lowest similarity score, the algorithm is able to find clusters that are very different from each other. An important advantage of this method is that it reduces the number of computations that needs to be computed. With this new anchor, the previously mentioned steps are applied recursively until all the sequences are assigned and we get a batch-wise dictionary of clusters, $Batch_D$. Throughout this process, the latent embeddings corresponding to these gesture-wise clusters are saved together. We refer the readers to Algorithm 1 in the appendix for more details.

*(2) Global Clustering:* After we obtain this batch-level dictionary of clusters, $Batch_D$, we update the global dictionary of clusters $Global_D$. For each of the batch clusters, we sample a sequence, $y_{samp}^b$, from the batch cluster $y^b$. Then, a sequence is sampled from each of the clusters in the global clusters $y^g$, we denote this as $y_{samp}^g$. We check whether $y_{samp}^b$ sequence belongs in an existing cluster in $Global_D$ with the same thresholding logic: $Sim(y_{samp}^b, y_{samp}^g) \geq \hat{\mu} + \hat{\sigma}$. If there exists a pair in that exceeds the threshold, we merge the batch cluster to the global cluster with the highest similarity value. Otherwise, we assign the batch cluster as a new global cluster in $Global_D$. Similarly to the batch clustering method, we save the corresponding latent embeddings in the global dictionary as well. We refer the readers Algorithm 2 for detailed description.

To tie this all back to our CC-NCE in Equation 7, we have $[y_{cb}^b, z_{cb}^b] \in Batch_D$ and $[y_{cg}^g, z_{cg}^g] \in Global_D$, where $cb$ indicates cluster index for the batch and $cg$ for the global dictionary. Given the i-th batch-level cluster, $[y_i^b, z_i^b]$, we treat the language embeddings, $z_i^b$, as the anchor sequences, because we want the language embeddings to learn the relationships present in the gesture space. Then, we find the most gesture-wise similar cluster in the global dictionary $y_i^+, z_i^+ = argmax \ (Sim(y_{cg}^g, y_i^b)), \forall \ [y_{cg}^g, z_{cg}^g] \in Global_D)$. We use the corresponding language embeddings of the most similar global cluster as the positive samples $z_i^+$. The language embedding sequences in other clusters in the global dictionary will be considered as negative samples $z_i^- = [Global_D \backslash z_i^+]$. With this assignment, we utilize properly assigned samples in our Crossmodal Cluster NCE in Equation 7.

## 5 EXPERIMENTAL SETUP

### 5.1 Dataset

We use the PATS dataset [1, 2, 14] as the benchmark to measure performance. It consists of aligned body poses, audio, and transcripts for 25 speakers. We choose five speakers (`maher`, `bee`, `lec_cosmic`, `oliver` and `colbert`) with a wide range of linguistic content and contrasting gesture styles for our experiments.

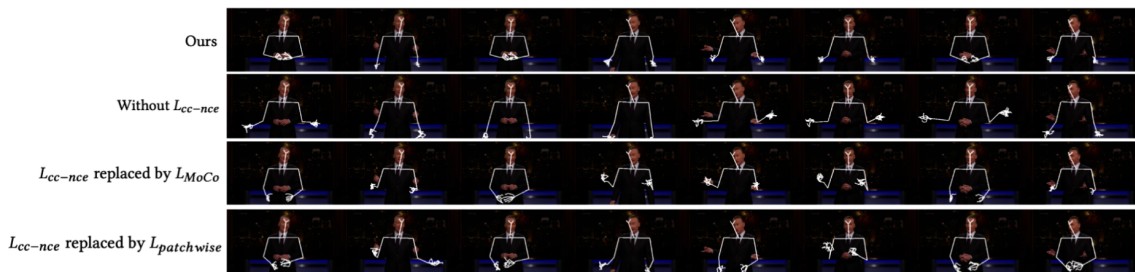

Fig. 4. Generated keypoints superimposed on ground truth images for easy comparison. The usage of contrastive learning produces gestures closer to the ground truth ($L_{MoCo}$, $L_{patchwise}$, $Ours$)

.

## 5.2 Baselines

We utilize the Multimodal Multi-Scale Transformer based GAN-architecture [1] as a primary building block of our proposed model. To the best of our knowledge, there have been no previous approaches that explicitly learn gesture-guided semantic spaces with contrastive loss functions. We compare our model with other self-supervised approaches, $L_{MoCo}$ and $L_{patchwise}$, by replacing the loss function $L_{cc-nce}$ in Equation 6.

$L_{cc-nce}$ **replaced by** $L_{MoCo}$: The contrastive learning proposed in MoCo [19] builds a large queue of data samples. The queue is referenced to find positive samples, if the encoded views are from the same image. Otherwise, the remaining elements are considered to be negative. This model is similar to our proposed $L_{cc-nce}$, without the utilization of clustering in the gesture space to assign positive and negative labels and relying on data augmentation and noise sampling for this assignment.

$L_{cc-nce}$ **replaced by** $L_{patchwise}$: Another contrastive learning approach: patch-wise contrastive learning [33] uses a specific contrastive loss, which maximizes the mutual information between the corresponding input and output patches. The mechanism aligns corresponding input-output patches at specific regions, which allows it discretize inputs into patches and use them as positives and negatives.

**Without** $L_{cc-nce}$: We also compare our proposed model without the $L_{cc-nce}$ loss function which boils down to the backbone model [1].

## 5.3 Experimental Methodology

In order to measure the precision and grounding of the generations, specifically relevance and timing of the gestures, we report the L1 distance between generated and ground-truth gestures. To measure the distribution in the gesture domain, we utilize the Fréchet Inception Distance (FID), which has been used in comparing gesture distributions [1, 44], which measures the distance between the distributions of the output generated poses and the ground truth. These results are included in the Appendix Table 2,

## 5.4 Implementation Details

The baselines were all trained with their respective hyperparameters. We remove the AISLE adapative reweighting mechanism in [1] for our backbone model as it feeds in various samples repeatedly into the model. Because our model constructs clusters in an online manner, the resampling method causes the clusters to be constructed with repeated

| Model | L1 ↓ | | | | | |
|---|---|---|---|---|---|---|
| Speaker: | maher | bee | lec_cosmic | oliver | colbert | Mean |
| Ours | **0.881 ± 0.02** | **0.918 ± 0.017** | **0.737 ± 0.032** | **0.777 ± 0.02** | 0.096 ± 0.007 | **0.682 ± 0.007** |
| Without $L_{cc-nce}$ [1] | 0.992 ± 0.024 | 0.955 ± 0.036 | 0.765 ± 0.046 | **0.775 ± 0.025** | 0.092 ± 0.004 | 0.716 ± 0.006 |
| $L_{cc-nce}$ replaced by $L_{MoCo}$ [19] | 0.983 ± 0.028 | 0.94 ± 0.058 | 0.763 ± 0.042 | 0.781 ± 0.021 | **0.091 ± 0.002** | 0.771 ± 0.086 |
| $L_{cc-nce}$ replaced by $L_{patchwise}$ [33] | 0.951 ± 0.033 | 0.937 ± 0.019 | **0.731 ± 0.019** | 0.874 ± 0.124 | 0.096 ± 0.003 | 0.769 ± 0.085 |

Table 1. Ablation of various contrastive loss mechanisms for 5 speakers in PATS in the task of generation of gestures in terms of precision (L1). *Ours* utilizes the proposed $L_{cc-nce}$ loss, whereas *Without $L_{cc-nce}$* utilizes no contrastive learning at all, as proposed in [1]. $L_{cc-nce}$ is replaced by two other contrastive learning mechanisms $L_{MoCo}$ [19] and $L_{patchwise}$ [33] for comparison.

samples, which can be problematic. Furthermore, in order to initialize the global mean and standard deviation of similarity scores for two pairwise sequences $\hat{\mu}$, $\hat{\sigma}$ for online self-supervised clustering, we iterate through the data for two epochs to find the mean $\hat{\mu}$ and standard deviation $\hat{\sigma}$ of the pairwise dot product similarity (referred to as *Sim*) of two arbitrary sequences of poses. During this time, the contrastive loss is not applied. Finally, the encoder in 4.1 which learns our gesture-aware embedding space is based on a U-Net structure [36].

## 6 RESULTS AND DISCUSSION

We substantiate our results by testing on five sampled speakers from the PATS dataset, displayed in Table 1. We give detailed metrics for each speaker for the precision metric L1 and the mean.

**Impact on Precision:** Our proposed model with the inclusion of CC-NCE produces better L1 scores than other baselines (Table 1). We see a significant decrease in L1 scores. This implies that our CC-NCE model produces better well-timed and relevant gestures compared to other baselines. Specifically, we see that other contrastive learning approaches, $L_{MoCo}$ and $L_{patchwise}$ have worse L1 scores than that of the bedrock model without any contrastive learning (Without $L_{cc-nce}$). This additionally shows that our proposed method of constructing clusters in the output domain and coercing the model to learn a pose-aware embedding space is beneficial.

**Impact on Coverage:** Although our results show improvements in precision, there are important limitations to consider. The qualitative figures shed insight to the trade-off between coverage and precision. We refer the readers to Table 2. We see our model having worse FID scores, which represents the coverage of the generated distribution. The no contrastive learning [1] method, which uses an adaptive importance sampling approach for better performance in coverage, produces the best results. We are providing additional incentive for the model to generate a limited subset of gestures, as we are mapping a large language space to a smaller subspace of gestures. Therefore, a decrease in the FID scores is explained by the trade-off between coverage and precision.

**Impact of $L_{cc-nce}$:** We demonstrate the effectiveness of our Crossmodal Cluster NCE Loss and display the resulting pose-aware embedding space in Figure 2. Firstly, the heatmap plots demonstrate that the self-supervised clustered pose sequences are indeed similar. Each row of the heatmap displays an overlay of three individual 64-frame sequences in a specific cluster (indices 6, 7, 9). The red color indicates movements in the right arm and the blue color represents that of the left arm. For cluster 7, the gesture is dominated by a raised right arm and an up and down motion of the left arm. For cluster 9, the speaker is at their rest pose, with slight up and down movements of the right arm. Finally, for cluster 6, we can see that the left arm is mainly static, with movements on the right arm. Visually, we can see that clusters 6 and 9 are quite similar, with movements mainly dominated by the right arm, whereas cluster 7 is quite different.

| Model | FID ↓ | | | | | |
|---|---|---|---|---|---|---|
| Speaker: | maher | bee | lec_cosmic | oliver | colbert | Mean |
| Ours | 48.52 ± 5.39 | 100.03 ± 20.74 | 44.43 ± 9.71 | 54.06 ± 9.38 | 5.85 ± 0.84 | 50.58 ± 7.15 |
| Without $L_{cc-nce}$ [1] | **21.38 ± 3.89** | **65.67 ± 11.35** | **23.14 ± 11.03** | **46.48 ± 1.12** | 6.77 ± 0.05 | **32.69 ± 3.90** |
| $L_{cc-nce}$ replaced by $L_{MoCo}$ [19] | 32.15 ± 20.83 | 74.892 ± 24.17 | 27.38 ± 16.71 | 48.78 ± 2.13 | 6.57 ± 0.16 | 39.66 ± 12.38 |
| $L_{cc-nce}$ replaced by $L_{patchwise}$ [33] | 26.45 ± 3.74 | 70.23 ± 10.52 | 38.95 ± 4.02 | 49.47 ± 9.47 | **5.48 ± 0.85** | 33.30 ± 3.74 |

Table 2. Ablation of various contrastive loss mechanisms for 5 speakers in PATS in the task of generation of gestures in terms of coverage (FID). *Ours* utilizes the proposed $L_{cc-nce}$ loss, whereas *Without $L_{cc-nce}$* utilizes no contrastive learning at all, as proposed in [1]. $L_{cc-nce}$ is replaced by two other contrastive learning mechanisms $L_{MoCo}$ [19] and $L_{patchwise}$ [33] for comparison.

In the pose-aware embedding space, we also see that clusters 6 and 9 lie in closer regions in the t-SNE plot of the language representations, in comparison to that of cluster 7. This demonstrates that the intra-cluster and inter-cluster relationships for gesture similarity and dissimilarity is indeed preserved in the latent space as well. If the clustering information was not effectively transferred to the latent space, we would not be able to visually see the clusters in the t-SNE plot located in similar regions.

**Qualitative Comparison:** We refer the readers to Figure 4, which shows a rendering of each model's generated gestures superimposed on the ground truth images for easy comparison of the quality of the generations. Our generated gestures are close to the ground truth. Specfically, the many-to-one grounding between a smaller subset of gestures and language allows for less noisy generations, which are confined to a smaller higher quality subset of gestures, which is due to the clustered gesture-aware embedding space. The bedrock model, denoted as "Without $L_{cc-nce}$" [1], whose model architecture is designed around minimizing the distribution difference between the generation and the ground truth, produces gestures that are quite diverse but nonetheless divulges from the ground truth. On the otherhand, the contrastive learning based methods Ours, $L_{MoCo}$ [19], and $L_{patchwise}$ [33], seem to generate more relevant and precision gestures, which shows higher levels of grounding.

**Limitations and Future Work** Certain speakers with greater diversity contain gesture sequences that are quite different from that of the majority of the cluster. The key challenge lies in constructing self-supervised clusters in both the temporal and spatial dimension. On the other hand, converting this into a supervised task, with annotations collected for gesture clusters, would make CC-NCE even more effective. Secondly, although the larger joint movements are natural, we observe that the generated gestures have finger keypoints that are abnormal for specific speakers. This may be due to the fact that the CC-NCE is confounding the final objective function, with the reconstruction loss, causing the output generations to be noisy, especially since finger keypoints in the data are noisy due to its versatile movements. Finally, excessive grounding information may contribute to mode collapse, as it encourages the model to produce similar subset of gestures. Studies need to be done to encourage grounding while preventing convergence to a smaller subset of modes.

## 7 CONCLUSION

In this paper, we studied crossmodal grounding in the context of many-to-one mapping between spoken language and gestures for the task of co-speech gesture generation. We introduced a new contrastive loss function Crossmodal Cluster NCE loss, which guides the latent space to learn the similarities and dissimilarities in the constructed clusters in the gesture domain. Furthermore, we offered a mechanism to cluster temporal sequences in an unsupervised and online

fashion. We demonstrated the effectiveness of this approach on a publicly available dataset, which indicated that our proposed methodology outperformed prior approaches in grounding gestures to language. We also observe, in-line with the precision-coverage trade-off, that encouraging higher precision degrades the coverage of the generated gestures.

This approach shows promise in a wide variety of crossmodal tasks to enforce stronger levels of grounding in a self-supervised manner, not specific to gesture generation. In addition, our Crossmodal Cluster NCE could be applied in a unimodal setting for a uni-modal self-supervised representation learning. Enforcing input modality representations to be able to distinguish similarities and dissimilarities within itself may be helpful where the input space is large. Furthermore, pertinent to our task of gesture generation, a more fine-grained clustering could be done spatially (clustering based on left arm/right arm movements separately) and temporally (considering differing levels of granularity). Finally, the relevance of the clusters to the domain can be amended by a domain-specific choice of similarity metrics such as DTW [10] for speed-invariant gestures.

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

## A APPENDIX

### A.1 Crossmodal Cluster NCE: Algorithmic details

In Algorithms 1 and 2 we describe, in detail, Batch Clustering and Global Clustering that are key components for estimating our proposed CC-NCE model.

---

**Algorithm 1** Recursive Batch Clustering

---

- $z^b$: is the encoded audio and language representation
- $y^b$: corresponding ground truth pose
- $L = torch.zeroes(|B|)$: vector to check if clustered
- $Batch_D = dict()$: dictionary for batch-wise clusters
- $\hat{\mu}, \hat{\sigma}$: mean and std. dev for similarity scores
- $Sim$: Similarity Function
- $C^b$ batch-wise cluster index

$a = rand(|B|)$
$C^b = 0$
**while** $L$ not all True **do**
  $C^b = C^b + 1$
  $L[a] = True$
  $y_a^b = y^b[a]$
  **for** $idx, score$ in $\texttt{enumerate}(Sim(y_a^b, y^b[\sim L]))$ **do**
    **if** $score \geq \hat{\mu} + \hat{\sigma}$ **then**
      $Batch_D[C^b]$ append $(y^b[idx], z^b[idx])$
      $L[idx] = True$
    **end if**
  **end for**
  $dissimseq, idx = TopK(sim, 1, largest = False)$
  $a = idx$
**end while**
return $Batch_D$

---

---

**Algorithm 2** Global Clustering

---

- $Batch_D$: dictionary for batch-wise clusters
- $Global_D$: dictionary for global clusters
- $C^g$: global cluster index
- $\hat{\mu}, \hat{\sigma}$: mean and std. dev for similarity scores
- $Sim$: Similarity Function

$y^g_{samp}$ = sample a pose sequence per cluster from $Global_D$

**for** $i, values$ in $Batch_D$ **do**

    $y^b_i, z^b_i = values$ ( contains aligned poses & embeddings)

    $y^b_{samp}$ = sample a single sequence from $y^b_{clus}$

    **for** $idx, score$ in enumerate($Sim(y^b_{samp}, y^g_{samp})$) **do**

        **if** $score \geq \hat{\mu} + \hat{\sigma}$ **then**

            $Global_D[idx]$ append $(y^b_{clus}, z^b_{clus})$

        **else**

            $C_g = C_g + 1$

            $Global_D[C_g + 1] = (y^b_{clus}, z^b_{clus})$

        **end if**

    **end for**

**end for**

return $Global_D$

---

