# OpenReview forum: "Crossmodal clustered contrastive learning: Grounding of spoken language to gesture"
_ACM.org/ICMI/2021/Workshop/GENEA — GENEA Workshop 2021 Oral_

### Official Review · Reviewer_Q8pw · 2021-07-13
**A useful proposition for modelling the speech-gesture relationship**

**Rating:** 9
**Confidence:** 4

**Review:**

-- Summary:
The paper presents a method for cross-modal grounding of speech and co-verbal gesture, to be used for gesture generation. The authors seek to address the many-to-one relationship of speech and gesture by using contrastive learning to train a language embedding space that clusters semantic markers close to each other if they are accompanied by the same/a similar gesture. Performance in compared to previous work and improvements are reported.

-- Strengths:
A novel a useful approach for modelling speech-gesture mapping. Well executed.

-- Notes:
It is hard to judge the quality of the results, a video would be very helpful.
A handful of things were not described clearly:
- It wasn't clear to me how the speech is represented when learning the language embedding; what exactly are the inputs to the encoder? How is the semantic information represented? More information is needed on the dimensions of input, embedding, and output dimensions.
- Lines 481-485: I'm not 100% sure what the procedure is here, could the authors clarify?
- Line 380: "first samples an anchor sequence" ... of language?
- Lines 388-389: "in the next section" actually refers to something in the same section, I believe ("Gesture-Based Clustering"). Having numbered sub-sections and using a numbered reference would be nice.
- When the term "crossmodal grounding" is introduced in the abstract, it is not immediately clear what this means.
- Ln. 694: "output generations for the to be noisy"
- Ln. 511/512: "we want the language embeddings to be learn the relationships"

---

### Official Review · Reviewer_cUrR · 2021-07-16
**A good workshop paper with contrastive loss for crossmodal clustering**

**Rating:** 6
**Confidence:** 4

**Review:**

This paper proposes a contrastive loss function to learn many-to-one mapping between gestures and spoken language in a self-supervised manner. Ablation studies and experimental results show that the proposed approach outperforms others in terms of the L1 score on the publicly available PATS dataset.

Strengths

+ Using a contrastive loss to learn a gesture-aware embedding space as proposed in the paper can be considered novel within this context.
+ Paper presents a comparison with appropriate baseline methods.

Weaknesses

Overall, this is a good workshop paper. However, there are two main issues.
- The method has been tested on one dataset only. It is not clear why this dataset is selected. It would be good to see further results on either the TED Gesture dataset or the GENEA challenge 2020 dataset. Looking at results on one dataset, conclusions remain partial.
- Experimental results have a couple of shortcomings. The proposed approach yields worse results in terms of FID. This discussion needs to be moved to the main paper and an in-depth analysis should be presented. Moreover, the main aim of this paper is to explicitly model many-to-one mapping between gestures and spoken language. However, Figure 4 shows the generated gestures only, and no visual results are presented to highlight the specific contribution of this paper.

Minor comments

- In figure 3, the first level of clustering is based on the pose. However, it is not clear how language and audio representation is obtained to learn the embedding space.
- Typos: “utiliizng”, “on prior. works”, “as a a new”, “to be learn”, “ground gestures”

---

### Decision · Program_Chairs · 2021-07-19

Accept (Oral)